# Different Nerve-Sparing Techniques during Radical Prostatectomy and Their Impact on Functional Outcomes

**DOI:** 10.3390/cancers14071601

**Published:** 2022-03-22

**Authors:** Iason Kyriazis, Theodoros Spinos, Arman Tsaturyan, Panagiotis Kallidonis, Jens Uwe Stolzenburg, Evangelos Liatsikos

**Affiliations:** 1Department of Urology, University of Patras, 26504 Patras, Greece; jkyriazis@gmail.com (I.K.); thspinos@otenet.gr (T.S.); tsaturyanarman@yahoo.com (A.T.); pkallidonis@yahoo.com (P.K.); 2Department of Urology, University Hospital of Leipzig, 04103 Leipzig, Germany; jens-uwe.stolzenburg@medizin.uni-leipzig.de; 3Department of Urology, Medical University of Vienna, 1090 Vienna, Austria; 4Institute for Urology and Reproductive Health, Sechenov University, 119435 Moscow, Russia

**Keywords:** prostate cancer, radical prostatectomy, nerve-sparing, techniques, functional, outcomes

## Abstract

**Simple Summary:**

Optimum preservation of potency and continence after radical prostatectomy (RP) are equally important surgical endpoints as cancer control itself. Nerve-sparing technique during RP has a major impact to both oncological and functional outcomes of the procedure and various different techniques have been developed aiming to optimize its outcomes. This literature review aims to summarize all different nerve-sparing techniques applied during RP from its first description from Patrick C. Walsh to its newer trends. The review underlines that optimum nerve-sparing expands far beyond recognising and preserving the anatomical integrity of the neurovascular bundles. It also emphasises that nerve-sparing is a field under constant development, with new technologies entering continuously the nerve-sparing field corresponding to the evolving open, laparoscopic and robotic-assisted RP approaches.

**Abstract:**

The purpose of this narrative review is to describe the different nerve-sparing techniques applied during radical prostatectomy and document their functional impact on postoperative outcomes. We performed a PubMed search of the literature using the keywords “nerve-sparing”, “techniques”, “prostatectomy” and “outcomes”. Other potentially eligible studies were retrieved using the reference list of the included studies. Nerve-sparing techniques can be distinguished based on the fascial planes of dissection (intrafascial, interfascial or extrafascial), the direction of dissection (retrograde or antegrade), the timing of the neurovascular bundle dissection off the prostate (early vs. late release), the use of cautery, the application of traction and the number of the neurovascular bundles which are preserved. Despite this rough categorisation, many techniques have been developed which cannot be integrated in one of the categories described above. Moreover, emerging technologies have entered the nerve-sparing field, making its future even more promising. Bilateral nerve-sparing of maximal extent, athermal dissection of the neurovascular bundles with avoidance of traction and utilization of the correct planes remain the basic principles for achieving optimum functional outcomes. Given that potency and continence outcomes after radical prostatectomy are multifactorial endpoints in addition to the difficulty in their postoperative assessment and the well-documented discrepancy existing in their definition, safe conclusions about the superiority of one technique over the other cannot be easily drawn. Further studies, comparing the different nerve-sparing techniques, are necessary.

## 1. Introduction

Prostate cancer is the second most commonly diagnosed cancer and the fifth leading cause of cancer death among men, with an estimated 1,414,259 new cases and 375,304 deaths worldwide in 2020 [1]. Radical prostatectomy (RP) is one of the gold standard treatment options for patients diagnosed with localized and locally advanced prostate cancer, who have a life expectancy greater than 10 years. Postoperative impairment of potency and continence are known adverse effects of RP. Regardless of the surgical approach which is applied (open, laparoscopic or robotic-assisted), preservation of the integrity of the neurovascular bundles (NVBs) is the key to maintain patients’ sexual function postoperatively. Furthermore, nerve-sparing (NS), along with other factors, has a positive impact on restoration of patients’ continence rates. Aiming to optimize its outcomes, NS technique has undergone numerous technical modifications throughout the years. In this article, we review the different NS techniques used in open, laparoscopic and robotic-assisted prostatectomy, emphasizing their impact on functional outcomes.

## 2. Anatomy of the NVBs

Historically, the nerves responsible for potency were first described in 1863 by Eckhard, C., when he defined nervi erigentus in animal models [2]. Nevertheless, the revolution in understanding the neuroanatomy of the prostate took place one century later, when, in a series of studies, Walsh, P. described the exact anatomy of the cavernous nerves and their role in ensuring potency after radical prostatectomy. According to Walsh, the NVBs are tubular structures consisting of the prostatic plexus of nerves along with vessels [3,4,5]. The location of NVBs in most males is posterolaterally and symmetrically to the prostate. Nevertheless, an anterolateral position or an asymmetrical posterolateral position on one side, and lateral on the other side, can be found in some cases [6]. They lie close to the tips of seminal vesicles and travel towards the posterolateral base of the prostate as an inferior extension to the pelvic plexus. From there, they run towards the apex of the prostate and membranous urethra enclosed within fascial planes. A triangular space which is delineated by three fascial planes (the prostatic fascia, the pelvic fascia and the Denonvillier’s fascia) surrounds the NVBs. This space is wider near the base of the prostate and narrower near the apex of the prostate [7].

In 2004, Costello et al. described the anterior and the posterior nerves of the NVBs. According to them, the posterior nerves consist the majority of NVB fibers and descend distally and dorsolaterally to the seminal vesicles, while the anterior nerves run near the posterior–lateral border of the seminal vesicles. The distance between the anterior and the posterior nerves is around 3 cm at the base of the prostate, coming closer to each other at mid-prostatic level (and thus forming a more well-defined bundle) and diverging again near the prostatic apex [8]. Tewari et al. revisited the anatomical foundations of the NVBs, describing a hammock-like arrangement of them in a trizonal distribution. The three zones consist of a proximal neurovascular plate (PNP), a predominant neurovascular bundle (PNB) and accessory neural pathways (ANPs) [9]. Menon et al. identified erectile nerves along the anterolateral aspect of the prostate and proposed a surgical technique that ensures the maintenance of these fibers (“Veil of Aphrodite” technique) [10]. Finally, additional nerve fibers were identified on the ventral aspect of the prostate [11]. A summary of the above anatomical description is presented in Figure 1.

## 3. Basic Principles of NS Techniques

NS techniques can be distinguished depending on the fascial planes which are dissected (intrafascial, interfascial or extrafascial), the direction used during dissection (antegrade, retrograde or a combination of two), the timing of the NVB release off the prostate (before or after manipulations on the prostatic vascular pedicles), the use of cautery (thermal or athermal), the application of traction (traction-free or non-traction-free techniques) and the number of the NVBs which are preserved (unilateral or bilateral NS). Transection, cautery, crush and traction are underlying mechanisms by which the cavernous nerves can be injured during manipulations, and many NS techniques have been developed in order to avoid one or more of them [12]. Despite this rough categorisation, several alternative NS techniques have been developed which cannot be integrated in one of the categories described above.

## 4. The Fascial Planes for NS

The prostate and the bladder are covered by a multilayer fascia called the endopelvic fascia, which is linked with both of them by collagen fibers. At the level of the prostate, the endopelvic fascia has an inner part overlying the prostate (prostatic fascia) and an outer part (Levator fascia or Lateral Pelvic Fascia). Posteriorly, the prostate and the seminal vesicles are covered by the Denonvilliers’ fascia, which laterally merges with the endopelvic fascia. Finally, a layer of fibromuscular smooth muscle located between the prostatic glandular units and the periprostatic connective tissue, along with vessels and nerves, constitute the prostatic capsule [13]. Based on the exact fascial plane of dissection, three different NS techniques can be described (Figure 2). In extrafascial NS, dissection is performed under the Denonvilliers’ fascia, as evidenced by the presence of perirectal fat on the posterior aspect of the dissection plane. In interfascial NS, the prostatic fascia is retained intact and dissection is performed within the plane between the prostatic fascia and the lateral pelvic fascia, which extends posteriorly as the plane between the prostatic fascia and the Denonvilliers’ fascia. Although the original NS procedure described by Walsh was performed using the interfascial technique, the discovery of supplementary nerve fibers on the anteriolateral surface of the prostate promoted the development of the intrafascial technique in an attempt to preserve them [14]. In intrafascial NS, dissection is performed within the plane between the prostatic capsule and the prostatic fascia. At the end of the intrafascial dissection, no additional tissue over the prostate is identified.

### 4.1. Intrafascial vs. Interfascial NS

Several studies have examined the impact of different fascial planes of NS on the functional outcomes of RP. Khoder et al. conducted a prospective study, in which 430 patients were included, comparing open complete intrafascial with interfascial RP. They concluded that the intrafascial prostatectomy offers better functional results in comparison to the interfascial approach, without compromising the oncological results a year after the procedure [15]. In another study including 147 patients, Potdevin et al. compared the outcomes of intrafascial versus interfascial NS in robotic-assisted radical prostatectomy (RARP). The authors observed a higher potency rate and a shortened time of return of continence following the intrafascial technique. Nevertheless, the positive surgical margin rates were higher in patients with pT3 disease in the intrafascial group [16]. Stolzenburg et al. compared the outcomes for interfascial and intrafascial NS endoscopic extraperitoneal RP, coming to the conclusion that the intrafascial technique is associated with significantly better potency in patients <55 years of age at 12 months and in patients 55–65 years of age at 6 and 12 months, with probably limited effect on the oncological outcomes. Moreover, the intrafascial technique was also associated with significantly improved continence rates at 3 and 6 months after surgery as compared with the interfascial approach [14]. Although, many studies report better outcomes following the intrafascial technique, the oncological safety of this procedure has been called into question. Curto et al. found a positive surgical margin rate of 30.7%, associated with the intrafascial approach, a rate much higher than that of other interfascial series [17]. In contrast, Wang et al. noted that the intrafascial NS, when cases are carefully selected, could offer an acceptable or, at least, equivalent positive surgical margin rate compared with the conventional interfascial approach [18]. In a recent meta-analysis, performed by Weng et al., the intrafascial technique has proven to be superior to the interfascial technique in terms of functional outcomes, likely due to lesser nerve damage. This study demonstrated better continence at 6 months and 36 months and better potency recovery at 6 months and 12 months postoperatively, associated with the intrafascial approach. Surprisingly, cancer control was also better with the intrafascial technique, possibly because patients in the interfascial group presented higher-risk cancer than patients in the intrafascial group [19]. The key steps of intrafascial NS are depicted in Figure 3.

### 4.2. Extrafascial vs. Non-Extrafascial (Intrafascial and Interfascial) NS

Shikanov et al. compared 110 cases of bilateral extrafascial NS versus 703 cases of interfascial NS. They observed significantly better sexual function in patients undergoing bilateral interfascial NS, while in lower-risk patients, bilateral interfascial NS did not result in significantly higher positive surgical margins rates [20]. Zhao et al. in a meta-analysis including 2096 patients from 7 eligible studies, compared the oncological and functional outcomes of intrafascial with non-intrafascial RP (including interfascial, extrafascial and no nerve-sparing approaches) in patients with low-risk localized prostate cancer. Meta-analysis demonstrated that the oncological outcomes were similar between the two groups, while the intrafascial approach was associated with lower postoperative complication rates, higher continence rates at 3 months and higher potency rates at 6 and 12 months following surgery, as compared to the non-intrafascial approaches [21].

## 5. Different NS Grading Systems

Various alternative grading systems have been proposed to describe the different degrees of NS. The intrafascial plane corresponds to the maximal extent of NS, while the extrafascial plane corresponds to the minimal extent. Montorsi et al. divided NS into full, partial and minimal as matching to intrafascial, interfascial and partial extrafascial dissections, respectively [22]. Tewari et al. proposed a new 4-degree stratification for preservation of the neurovascular bundles, using the veins which are situated on the lateral aspect of the prostate as landmark. According to them, grade 1 matches to a complete intrafascial dissection and is a dissection between the periprostatic veins and the pseudocapsule of the prostate. Grade 2 matches to an interfascial dissection and is a dissection performed just over the veins, while grade 3 dissection leaves more tissue over the veins and the prostate. Finally, grade 4 matches to an extrafascial dissection [23]. Schatloff et al. proposed a 5-degree stratification for the definition of the dissection planes, using the “landmark artery” (LA), which runs on the lateral aspect of the prostate, as a reference point. The LA can be recognized, intraoperatively, in up to 73% of cases. A grade 5 dissection, during which the prostate and the NVBs can be separated without the need of sharp dissection, matches to a maximal NS (complete intrafascial dissection) and is performed medially to the LA just outside the prostatic fascia. A grade 4 dissection is performed using sharp dissection in a plane between the LA and the prostatic pseudocapsule across the NVB, while in a grade 3 dissection, the plane of NS is created laterally to the LA, and thus the artery is clipped at the level of the prostate pedicle. Finally, for a grade 2 dissection, NS is performed several millimetres laterally to the artery, while for a grade 1 dissection, an extrafascial dissection is performed and the NVBs are not spared (Figure 1) [24]. A summary of the different grading systems is demonstrated in Table 1.

## 6. The Direction of NS

The conventional open retropubic RP, proposed by Walsh, includes a completely retrograde NS dissection. The direction of dissection is from the prostatic apex towards the base, while the vascular pedicles are taken last after the dissection of the NVBs. A similar retrograde approach has also been described in laparoscopic and robotic surgery. The retrograde approach has the advantage of early recognition and release of the NVBs from the prostate, before controlling the posterior pedicle [7,12]. Patel et al. described their technique of early retrograde release of the NVBs during RARP in an athermal way with minimisation of traction. In order to perform this approach, the LA must first be recognised in the lateral aspect of the prostate. A plane is developed between the LA and the prostate, which is continued posteriorly, and then the dissection proceeds in a retrograde way towards the base of the prostate (Figure 4). Prostatic pedicles are clipped last, resulting in a natural traction-free release of the NVBs off the prostate. Describing their results in 397 patients, they reported that 87.7% of patients with Sexual Health Inventory for Men (SHIM) >21, and 73% with SHIM between 17 and 21 preoperatively, were potent with or without the use of phosphodiesterase-5 inhibitors, after a 3-month follow-up [7,13,24].

Conversely, in the antegrade approach, the direction of the dissection is from the prostatic base towards the apex with the vascular pedicles being transected first [7,12]. Antegrade NS is a widely used practice among laparoscopic and robotic surgeons. The procedure starts with a gentle upward traction of vas and seminal vesicles in order to reveal the prostatic pedicles. Counter traction of the prostate exposes the triangular space between the lateral pelvic fascia, the Denonvilliers’ fascia and the prostatic fascia and either the interfascial or the intrafascial dissection is performed [7,24]. An antegrade approach has been described in open surgery as well [25,26]. According to Carini et al., open antegrade NS constitutes a less challenging procedure with similar results to those reported by the retrograde approach [25]. Finally, a partial retrograde approach has been described which preserves the advantages of the antegrade NS, but takes the vascular pedicles last as in the retrograde NS [12].

Regarding the impact of different directions of NS on functional outcomes, a questionnaire-based assessment demonstrated that 67% of patients undergoing retrograde, and 76% of patients undergoing antegrade NS laparoscopic RP, were able to engage in sexual intercourse (with or without phosphodiesterase-5 inhibitors) postoperatively [27]. On the contrary, in a nonrandomized comparative study, Ko et al. reported that, in patients with normal preoperative erectile function, a retrograde direction of nerve-sparing during RARP was associated with significantly higher potency rates at 3, 6 and 9 months compared with an antegrade direction of NS, without compromising cancer control [28].

## 7. Unilateral versus Bilateral NS

There is a marked controversy in the outcomes of studies examining the impact of unilateral versus bilateral NVBs preservation. Finley et al. in a study including 96 patients subjected to RARP, failed to demonstrate significant difference in outcomes between unilateral and bilateral NS. Still, the study was limited by the fact that it was designed to compare cautery versus cautery-free NS and not unilateral vs. bilateral NS [29]. Similarly, Nilsson et al. documented that the majority of preoperatively potent patients subjected to RARP were potent at 1-year follow-up irrespective of whether they had a bilateral, unilateral or semi-NS approach [30]. On the other hand, according to Greco et al., patients undergoing bilateral NS had higher rates of potency than patients undergoing unilateral NS after an intrafascial NS laparoscopic RP procedure [31]. In addition, Avulova et al. performed a population-based prospective observational study (the CEASAR study) in which they observed better potency and continence outcomes after bilateral NS as compared to unilateral NS [32].

## 8. The Functional Impact of Using Energy and Nerve Traction during NS RP

The potential injury and functional recovery of the nerves depends first of all on the nature of the nerves. While some studies have failed to identify myelinated nerve fibers originating from inferior hypogastric plexus [33], others have shown both myelinated and unmyelinated components in prostatic nerves [34,35]. There are several classifications used for evaluating and terming nerve injuries [36]. Based on Seddon’s classification, the nerve injuries can be divided into 3 types: neurapraxia, axonotmesis and neurotmesis [37]. The neurapraxia is the first-degree injury, commonly caused by mechanic blunt trauma to the nerves. The recovery after this injury may take as long as 12 weeks. Axonotmesis, the second-degree injury, is termed so due to axonol injury, yet preserving the surrounding connective tissue. Depending on the injury distance and with the axonol growth rate of 1 mm/day delayed nerve recovery, up to 24 months may be required. Neurotmesis is the most severe injury of the nerve commonly resulting in an irreversible loss of nerve function [38].

The nerves of NVBs, are sensitive to thermal energy which is diffused during current use in adjacent structures [13]. Suspecting that thermal injury of NVBs could be responsible for the postoperative loss of potency, many investigators tried to develop totally athermal methods. Theoretically, cautery-free techniques manage adequate hemostasis, while improving the return of erectile function by minimising injury to the NVBs. Vascular clips, suture ligation and bulldog clamps with suturing constitute the most common cautery-free techniques, used during prostatic vascular pedicle manipulations [12]. In addition, several hemostatic agents have been used for the purpose of controlling bleeding. Although hemostatic agents were used in Ahlering’s initial report, their inability to manage hemostasis in 15% of the cases led to their substitution by suture ligation [39]. Gill et al. also reported problems with FloSealTM in their laparoscopic RP series, frequently requiring secondary clip placement [40]. Moreover, usage of bioadhesives near the NVBs can potentially injure them, as a result of a lymphocytic inflammatory reaction and fibrosis [12].

The impact of different hemostatic energy sources to the integrity of NVBs was initially documented in the experimental setting. Ong et al. evaluated cavernous nerve function on 12 dogs, which were divided into 4 groups, each subjected to NS using conventional dissection with suture ligatures, monopolar electrosurgery, bipolar electrosurgery or ultrasonic shears, respectively. The authors observed that use of energy sources near the NVBs was associated with a considerable decrease in erectile response both acutely and after 2 weeks; while following conventional dissection with suture ligatures, the erectile response to nerve stimulation was unaffected [41]. It has been also shown that monopolar and bipolar energy have the almost similar risk of heat generation and potential tissue injury [42]. However, a decreased risk of nerve injury with bipolar energy can be observed when cut and catherization is performed, so-called touch cautery, due to preservation of adjacent blood flow [43].

In accordance with these findings, several clinical studies confirmed the importance of athermal dissection. Ahlering et al. described a clipless cautery-free approach for NS, using bulldog vascular clamps and sutures, and reported a nearly 5-fold rate of improvement in potency recovery as compared to a group where NS took place using cautery. Defining potency as “erections hard enough for vaginal penetration with or without the use of PDE-5 inhibitors” in the cautery group, 14.7% of patients were potent after 9 months and 63.2% after 24 months, as compared to 69.8% (after 9 months) and 92% (after 24 months) for the cautery-free group [39,44,45]. Likewise, Chien et al. described analogous findings during a completely athermal RARP procedure, reporting a faster return and preservation of sexual function. In their modified clipless antegrade NS technique, after developing the posterior plane of the prostate towards the apex in the midline, they released the vascular pedicles and the NVBs in a medial-to-lateral direction using a combination of sharp and cold scissors. They only used judiciously bipolar cautery, avoiding clips and monopolar cautery. According to them, the potency rates after this approach, using a 36-item health survey questionnaire, were 47%, 54%, 66% and 69% after 1, 3, 6 and 12 months, respectively [46]. Gill et al. using real-time Doppler transrectal ultrasound, showed that bulldog clamping of the lateral vascular pedicles was associated with preservation of blood flow in the NVBs and restriction of the need for cautery [40]. Finally, Fagin et al. compared three different NS techniques (selective bipolar cautery, an athermal “clip and peel” posterior dissection technique and an athermal combined anterior and posterior dissection technique with clips and sharp dissection). The authors reported better recovery of potency with the athermal techniques, with the combined anterior and posterior approach being the superior of the two athermal techniques. This approach was also associated with the lowest positive margin rates [47].

Regarding the impact of different energy sources on functional outcomes, the available data are limited. Pagliarulo et al. compared the athermal and the ultrasonic NS laparoscopic RP procedures, coming to the conclusion that the use of an ultrasonic device did not have a negative impact on long-term potency and continence outcomes, nor did it lead to early biochemical recurrence, as compared to the athermal approach [48]. Pastore et al. performed a prospective randomized study, comparing radiofrequency and ultrasound scalpels on functional outcomes of laparoscopic RP and documented that the radiofrequency scalpel was associated with better functional outcomes [49].

With regard to the impact of traction on the integrity of NVBs, many studies have documented the positive effects of traction-free techniques. Kowalczyk et al. observing 610 patients who underwent RARP, 342 of whom were with avoidance of countertraction of the NVBs during NS, reported earlier sexual function recovery in the traction-free group (45% versus 28% at five months). However, potency rates were the same among the two groups at 1 year [50]. Similar results were reported by Masterson et al., who modified their technique in order to avoid countertraction of the NVBs during RARP and observed improved rates of erectile function recovery during a six-month period [51]. Finally, Mattei et al. presented the results of their lateral approach for the interfascial dissection of the NVBs without tension and any use of electrocautery. In their study, one week after catheter removal, 80% of patients had complete early urinary continence and a high rate of patients reported spontaneous erections or penile tumescence, while at the 4-month follow-up visit, 92.4% of patients were completely continent and 65% of patients were considered potent [52].

## 9. “Veil of Aphrodite” and “Super Veil” Technique

The “Veil of Aphrodite” (also called the curtain dissection) is a modified NS technique during which the prostatic fascia is detached off the prostate and remains as a supportive structure over the ipsilateral NVB. After this approach, the periprostatic tissue along with the NVBs hung like a curtain from the pubourethral ligaments [53]. For this purpose, the interfascial plane between the posterior prostatic fascia and the Denonvilliers’ fascia is extended towards the apex, between 1 and 5 o’clock for the right side and between 6 and 11 o’clock for the left side. The pedicles are then divided and the prostatic fascia is incised anteriorly so as to enter the intrafascial plane [13]. Except for the preservation of cavernous nerves in the lateral prostatic surface, better functional outcomes following the “Veil” technique can be explained by the reduced traction injury to the posterolateral NVBs during a precise dissection on the lateral surface of the prostate, resulting in reduced neurapraxia of the cavernous nerves. Furthermore, during this technique, the dissection is following avascular planes requiring little thermal energy [54]. Menon et al. reported the potency outcomes on 1142 patients undergoing RP with this particular technique (out of 2652 patients undergoing RARP at their institute). According to them, after 1-year follow-up, 70% of preoperatively potent patients undergoing bilateral NS, were also potent after the surgery with or without the use of phosphodiesterase 5 inhibitors [55]. In addition to the conventional “Veil” technique, the recognition that 25% of the NVBs can be found on the anterior surface of the prostate led Menon et al. to modify their technique, in an attempt to preserve the pubovesical ligaments and the dorsal venous complex (DVC) intact. During this approach, called the “Super Veil” technique (or “Super Veil” sparing), the interfascial dissection is extended more anteriorly between the 11 o’clock and 1 o’clock positions (Figure 5). As reported by Menon et al., out of 85 patients undergoing “Super Veil” NS RARP, who attempted sexual intercourse, 94% had an erection strong enough for penetration at a median follow-up of 18 months. Comparing erectile function in patients undergoing the “Veil” and the “Super Veil” technique, the authors came to the conclusion that, while average SHIM scores were similar between the two groups, patients undergoing the “Super Veil” technique were able to achieve intercourse earlier than the “Veil” patients [56].

## 10. Hydrodissection of the NVBs

Guru et al. described an athermal NS technique, using 1:10,000 epinephrine solution diluted in 0.9% normal saline, which is injected into the lateral prostatic pedicle with an injection cannula needle. As reported by them, in a series of 10 potent patients who underwent bilateral NS RARP with hydrodissection, intraoperative parameters were favorable. Nevertheless, no long-term potency outcomes of the technique have been reported [57].

## 11. The Role of Lasers in NS

The ability of CO_2_ laser to deliver focused ablation, while minimising thermal spread to surrounding tissues, prompted Cheetham et al. to try the application of a flexible carbon dioxide laser fiber for dissection of the NVBs during RARP. Using OmniGuide BeamPath URO-LG CO_2_ laser fiber for dissection, bilateral lateral fascial antegrade NS was performed in 10 patients. After clipping and dissection of the prostatic pedicles, the laser fiber was used in order to develop the plane between the prostate and the NVBs. The authors reported that the laser system enabled a meticulous dissection of the NVBs and easy fascial layer identification. According to them, 90% of patients who underwent this approach had gained urinary continence at 3 months after surgery with no data concerning potency reported [58]. The 532 nm KTP laser is characterised by both good hemostatic properties and a shallow depth of penetration. Gianduzzo et al. using a canine model, compared the impact of three different techniques (KTP laser, ultrasonic shears and athermal technique with clips and cold scissors) on cavernous nerve function after laparoscopic unilateral NVB mobilisation. Performing a thermographic study, they reported significantly lower collateral thermal damage in the KTP laser group in comparison with the ultrasonic shears group. Thus, they concluded that, concerning cavernous nerves’ function preservation, the KTP laser was comparable to the athermal technique and superior to the ultrasonic shears [59].

## 12. Retzius-Sparing RARP

A Retzius-sparing approach for robot-assisted laparoscopic prostatectomy was first reported by Galfano et al. Retzius-sparing RARP is not a distinct NS approach, but rather a totally different access of the prostate, which is performed via a transperitoneal posterior approach, preserving the neurovascular bundles, the pudendal arteries and supporting structures of the prostate which are located anteriorly to it (such as the puboprostatic ligament) [60]. Although the Retzius-sparing technique is associated with early acquisition of urinary continence [61,62,63], the recovery rate of potency after this procedure is similar with that after conventional nerve-sparing RARP in many studies [62,64,65,66].

## 13. Neurovascular Structure Adjacent Frozen Section Examination (NeuroSAFE) in NS

The NeuroSAFE technique has been introduced in order to balance the preservation of the NVBs along with the incidence of positive surgical margins. When there are concerns that prostate cancer may extend, beyond its capsule into the NVBs, NS increases the risk of positive surgical margins, while in the case of cancer infiltration, the ipsilateral NVB can be sacrificed. During the NeuroSAFE technique, intraoperative fresh-frozen section analysis of the posterolateral aspect of the prostate margin is performed in the pathology department, in order to assess whether cancer extends beyond the capsule. In the case of a positive result from the pathologist, the ipsilateral NVB is resected along with the rectolateral part of the Denonvilliers’ fascia. Schlomm et al. examined the role of NeuroSAFE procedure in both open and robotic NS RPs in a cohort of 11,069 patients undergoing RP, 5392 (48.7%) of which were carried out with the NeuroSAFE approach. The authors found that the NeuroSAFE approach increased the NS frequency, while simultaneously decreased the positive surgical margins rate [67]. Although this is a retrospective observational study, Dinneen et al. are currently performing a multi-centre, single-blinded randomised controlled trial (NeuroSAFE PROOF RCT) in order to evaluate the efficacy of this method [68].

## 14. The Role of Near-Infrared Fluorescence (NIRF) and Indocyanine Green (ICG)

The role of the landmark artery (LA) in NS has been described in a previous section of this article. However, identification of this artery can be challenging especially for a novice urologist or in the case of a patient with anatomical variations [69]. Kumar et al. proposed the use of near-infrared fluorescence (NIRF) and indocyanine green (ICG) for identifying the LA and performed a prospective case series examining the usefulness of this technology during ten cases of RARP. According to the authors, the LA and its course were identified in 85% of cases, without causing any complications or increasing the operating time [70].

## 15. Bioengineering in NS

Even in the most well-preserved NVBs, after a NS procedure a minimal nerve injury takes place, resulting in neurapraxia. In an effort to promote regeneration of cavernous nerves to improve functional outcomes, many neurotrophic factors have been studied [7,69]. Patel et al. performed a propensity-matched analysis, assessing the role of dehydrated human amnion/chorion membrane (dHACM) wrap, which was placed bilaterally around fully preserved NVBs after the vesico-urethral anastomosis. For this purpose, 58 preoperatively potent and continent patients, undergoing NS RARP, were included. According to the authors, potency at 8 weeks returned in 65.5% of patients in the dHACM group and 51.7% of patients in the non-dHACM group, with the mean time to potency being shorter in the dHACM group (1.34 months vs. 3.39 months). Moreover, SHIM scores were also higher for the dHACM group (mean score 16.2 vs. 9.1). They thus concluded that dHACM wrap around the NVBs promotes early return to potency, without increasing the operative time and without leading to blood loss or negative oncologic outcomes [71]. Similar outcomes were demonstrated by Ogaya-Pinies et al. in a series of 940 patients [72]. Porpiglia et al. studied the 1-year efficacy of the application of chitosan membrane (ChiMe), another biomembrane with neurotrophic factors, on the NVBs during NS RARP. They reported significantly improved potency rates at 1 and 2 months following surgery in the ChiMe group as compared to the control group (36.76% vs. 25.88% and 52.2% vs. 39.22%, respectively). The potency rates were also higher beyond 2 months for the ChiMe group, with that difference being not statistically significant, though [73]. Finally, Hinata et al. investigated the efficacy of the application of a hyaluronic acid-carboxymethyl cellulose membrane (HA/CMC), on the prostatic bed and neurovascular plate during NS RARP, concluding that this approach significantly shortened the duration of postoperative incontinence after both unilateral and bilateral NS RARP [74].

## 16. Devices for Identifying the Cavernous Nerves Intraoperatively

Identification of the exact location and course of the cavernous nerves intraoperatively facilitates their preservation, leading to better functional outcomes postoperatively. In that effort, many innovative methods of nerve-tracing have been developed. To begin with, electrical nerve stimulation devices can be used and the erectile response can be evaluated both indirectly by physiological measurements, such as penis tumescence, and directly by measuring the neuroelectrical activity in the corpus cavernosum. Moreover, fluorescence imaging can precisely detect the cavernous nerves, applying fluorescein or indocyanine green, either by local injection into the cavernous nerves by way of the penis or by systemic injection intravenously [75]. Finally, transrectal ultrasonography and laparoscopic Doppler ultrasound (LDU) probe can both help identification of these nerves [69,75]. A detailed description of these methods exceeds the purposes of this article.

## 17. Discussion

The outcomes after a RP procedure have been traditionally reported as “trifecta”, including urinary continence, potency and cancer control. Patel et al. proposed the addition of perioperative complications and surgical margin status to the “trifecta” outcomes, describing the “pentafecta” outcomes, which, according to them, more precisely reflect patients’ expectations after a RP procedure [76]. Postoperative potency and continence rates are used in order to assess the functional efficacy of a RP procedure. Nevertheless, it is extremely difficult to accurately predict the outcomes after RP, despite the technique used, because they depend on many factors, and thus, to a certain degree of subjectivity. For example, the potency rates can differ depending on pre-operative erectile function, patient comorbidities, type and extent of NS, patient’s age, frequency of intercourse, use of medications and surgeon’s experience [13]. Similarly, preservation of continence and fluctuations in urinary incontinence rates after a RP procedure are multifactorial, depending not only on anatomical and surgical aspects (such as careful dissection and meticulous anatomical reconstruction), but on patients’ characteristics too [77]. In order to thus compare the different NS techniques and assess their outcomes, all these factors should be taken into consideration. It is extremely important to compare patients with the same characteristics in every case. Furthermore, there is a huge discrepancy in the definition of potency and continence after NS RP. Different studies should use the same definitions, as much as possible, so that comparable results are reported. Finally, many studies included in this article are comparative studies, lacking statistical significance. The implementation of more randomised clinical trials, systematic reviews and meta-analyses in the field of different nerve-sparing techniques is of vital importance.

## 18. Conclusions

NS during RP is the key to achieve optimal functional outcomes after surgery. Its introduction from Walsh, its worldwide adaptation and its further refinement by modern urology techniques enabled many patients, diagnosed with prostate cancer, to maintain a normal quality of life after RP. Bilateral nerve-sparing of maximal extent, athermal dissection of the neurovascular bundles with avoidance of traction and utilization of the correct planes remain the basic principles for achieving optimum functional outcomes. Given that potency and continence outcomes after radical prostatectomy are multifactorial endpoints in addition to the difficulty in their postoperative assessment and the well-documented discrepancy existing in their definition, safe conclusions about the superiority of one technique over the other cannot be easily drawn. Further studies, comparing the different nerve-sparing techniques, are necessary.

## Figures and Tables

**Figure 1 cancers-14-01601-f001:**
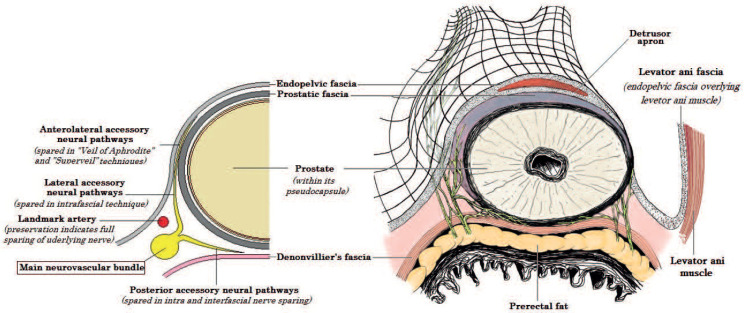
Periprostatic fascias and their relation to neurovascular bundles including the main neurovascular bundle, the posterior, the lateral and the anterolateral accessory neural pathways. Surgical techniques developed to protect each element of the neurovascular pathway are highlighted.

**Figure 2 cancers-14-01601-f002:**
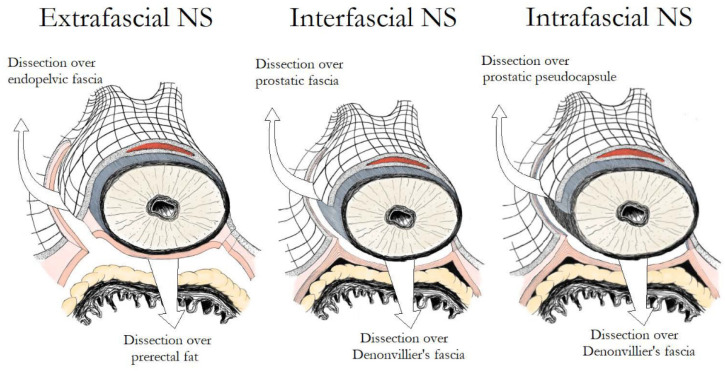
Main differences in fascial development between extrafascial, interfascial and intrafascial nerve-sparing.

**Figure 3 cancers-14-01601-f003:**
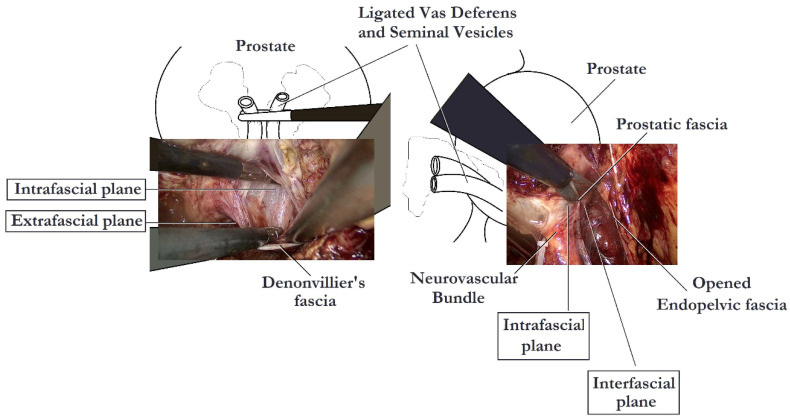
Nerve-sparing intrafascial laparoscopic radical prostatectomy.

**Figure 4 cancers-14-01601-f004:**
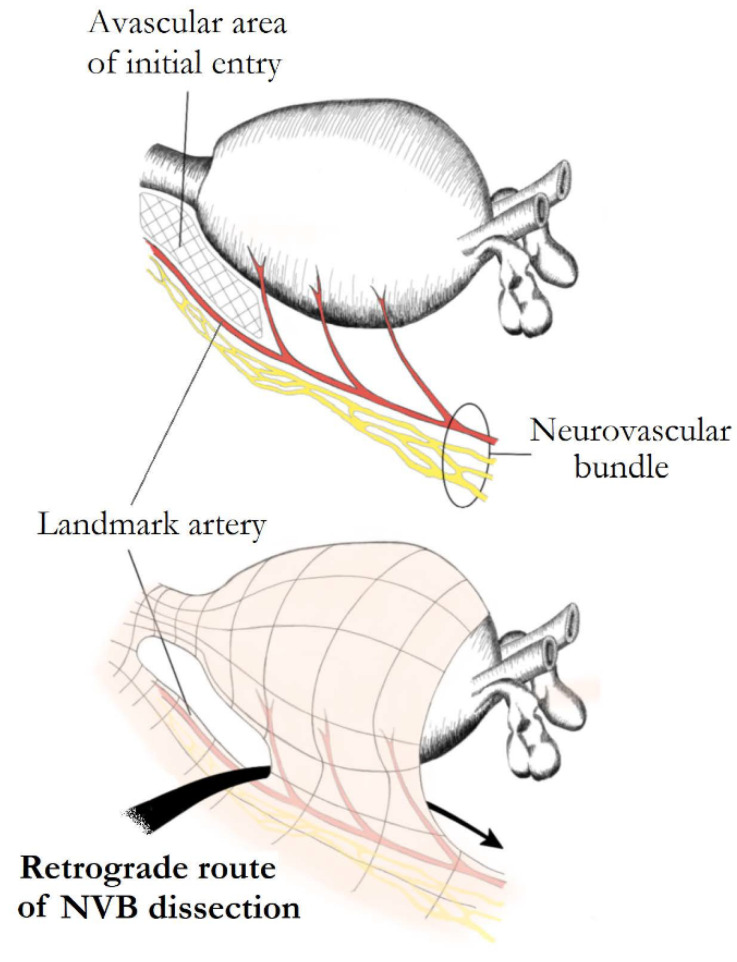
Retrograde nerve-sparing. After posterior prostatic dissection, an avascular area at the lateral margins of the prostatic apex overlying the landmark artery (LA) of neurovascular bundles is identified and used as the initial point of dissection. The bundles are further dissected in a retrograde fashion. Preservation of the LA ensures a complete preservation of the ipsilateral main neural branch.

**Figure 5 cancers-14-01601-f005:**
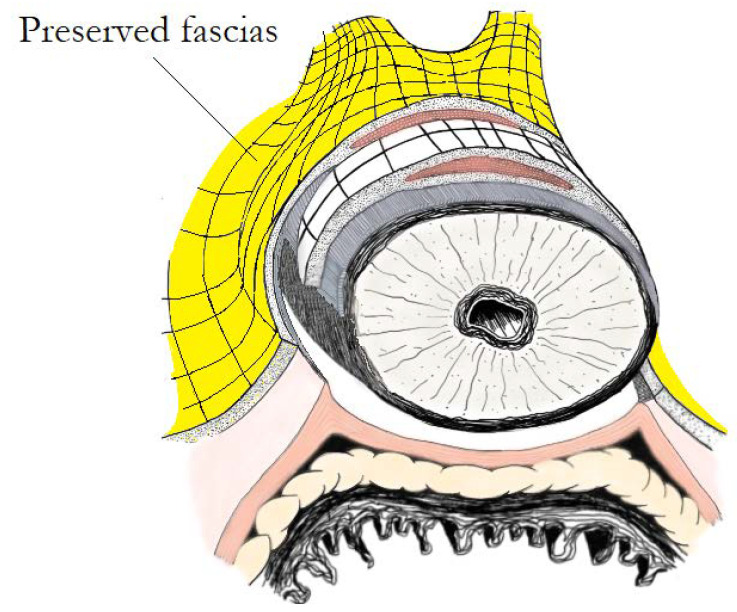
Super Veil of Aphrodite technique: a fascial plane between prostatic pseudocapsule and anterolateral periprostatic tissue is developed in the apical area. Preserved fascias (including part of puboprostatic ligaments-detrusor apron and part of anterolateral endopelvic and prostatic fascias) are highlighted in yellow.

**Table 1 cancers-14-01601-t001:** Different nerve-sparing grading systems.

Nerve-Sparing Grading System	References	Anatomical Landmark	Number of Grades	Description of Different Grades
**Fascial planes**	Stolzenburg et al. [14]	Periprostatic fasciae	3	Intrafascial Interfascial Extrafascial
**Extent of nerve-sparing**	Montorsi et al. [22]	Neurovascular bundles	3	Full nerve-sparing (matches to an intrafascial dissection) Partial nerve-sparing (matches to an interfascial dissection) Minimal nerve-sparing (matches to a partial extrafascial dissection)
**4-degree approach for preservation of the neurovascular bundles**	Tewari et al. [23]	The veins which are situated on the lateral aspect of the prostate	4	Grade 1 (matches to a complete intrafascial dissection) Grade 2 (matches to an interfascial dissection) Grade 3 Grade 4 (matches to an extrafascial dissection)
**5-degree approach for the definition of the dissection planes**	Schatloff et al. [24]	The “landmark artery” (LA), which runs on the lateral aspect of the prostate	5	Grade 1 (matches to an extrafascial dissection) Grade 2 Grade 3 Grade 4 Grade 5 (matches to a complete intrafascial dissection)

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
