# Peer review of "Different Nerve-Sparing Techniques during Radical Prostatectomy and Their Impact on Functional Outcomes"

_cancers, 2022, doi:10.3390/cancers14071601_

Round 1

Reviewer 1 Report

Different nerve-sparing techniques during radical prostatectomy and their impact on functional outcomes

This is a comprehensive and very well written paper. A few comments.

Comments to authors-

  1. No matter how good a procedure, nerve sparing success will have no effect on men with existing ED and if hidden EF co-morbidities. The authors should include some discussion of factors that impact recovery of sexual function including primarily age, preoperative IIEF-5 score, and co-morbidities (especially hypertension) and free testosterone levels.
  2. As the authors have nicely introduced Basic principles of NS techniques – they need to have a similar discussion regarding nerve anatomy and nerve injury as discussed below.
  3. Starting at line 253 the paper suggests that the injured nerves are unmyelinated as referenced Patel and colleagues [13]. This section needs reconsideration. This is a critical issue. There is no evidence that the NVBs are unmyelinated as they course along the prostate. All anatomic descriptions of the NVB indicate clearly that it is myelinated. The NVBs as described in Grey’s Anatomy (and all legitimate anatomy texts) are parasympathetic which is a system characterized by a “long myelinated preganglionic nerve” that exits the Sacral spinal cord and passes along the prostate down to the cavernosal bodies where they connect with a second ganglion which then innervates the corporal bodies with small and short 2-4mm postganglionic nerves which are unmyelinated. Unmyelinated (post ganglionic) nerves in the penis obviously are not injured during a RP.
  4. Further, the physiology of nerve recovery timeline is clearly supported by the anatomic descriptions of the long pre-ganglionic myelinated nerve taking 1-2 years to recovery. This of course explains why what appears to be a perfectly preserved nerve after prostatectomy takes months to 2 years to recover. Further, it is also supported by Seddon’s model of nerve injury and recovery and Wallerian degeneration principles.
  5. The authors need to include a clear description of nerve recovery as described by Seddon in the 1940s. There continues to be an embarrassing lack of physiologic knowledge throughout medicine. There is no such word as Neuropraxia. As described by Seddon the word is Neurapraxia (except now as it has mistakenly been “colloquially” accepted for the correct term).
  6. And neurapraxia and more important axonotmesis trauma only occurs in myelinated nerves. Myelinated nerves can suffer these two types of recoverable injury both of which require that the myelinated sheath is intact, i.e. good surgical preservation of the NVBs. Neurapraxic injury is like a concussion and recovers within weeks. The more common and typical recoverable injury is axonotmesis where the (axon) nerve is injured (by heat or traction along the prostate) and then goes through Wallerian degeneration and as long as the sheath was preserved, the axon will grow back through the sheath to the cavernosal bodies at about 1 mm/day and hence the delayed recovery of 9-24 months.
  7. In the presentation of thermal injury, there is strong evidence of another school of thought that should be added. Several papers point out that not all thermal injury is the same. In 2002-4 thermal injury occurred when the prostatic vascular pedicles were controlled with bipolar cautery paddles. There are papers and physics that indicate the 20-60 watts of monopolar or bipolar create the same amount of heat energy and potential injury. The major difference between the two is that bipolar paddles when applied to the pedicles occludes blood flow creating desiccating heat and injury. This point was published by Khan F, et.al. J. Endo 2007:21,1195-8.  DOI: 10.1089/end.2007.9908   Subsequently, cut and cauterizing which, unlike the bipolar paddles, preserves adjacent blood flow and has been demonstrated to have the same outcomes as an athermal approach.   Hoffman M, et.al. Eur. Urol. 81(2022),104-09. https://doi.org/10.1016/j.eururo.2021.07.005   

Author Response

  1. No matter how good a procedure, nerve sparing success will have no effect on men with existing ED and if hidden EF co-morbidities. The authors should include some discussion of factors that impact recovery of sexual function including primarily age, preoperative IIEF-5 score, and co-morbidities (especially hypertension) and free testosterone levels.

Authors’ response: The authors thank the reviewer for the comment. There is a short discussion in the “discussion” section stating the potential risk factors for post prostatectomy erectile dysfunction (lines 487-489). With respect to the reviewer’s comment, the aim of our article was to present and discuss the effect of different nerve sparing techniques. A detailed discussion of each single risk factor was not in the scope of the current article and thus, we believe, it should not be amended.

  1. As the authors have nicely introduced Basic principles of NS techniques – they need to have a similar discussion regarding nerve anatomy and nerve injury as discussed below.

Authors’ response: The authors thank the reviewer for the feedback and kind suggestion.

  1. Starting at line 253 the paper suggests that the injured nerves are unmyelinated as referenced Patel and colleagues [13]. This section needs reconsideration. This is a critical issue. There is no evidence that the NVBs are unmyelinated as they course along the prostate. All anatomic descriptions of the NVB indicate clearly that it is myelinated. The NVBs as described in Grey’s Anatomy (and all legitimate anatomy texts) are parasympathetic which is a system characterized by a “long myelinated preganglionic nerve” that exits the Sacral spinal cord and passes along the prostate down to the cavernosal bodies where they connect with a second ganglion which then innervates the corporal bodies with small and short 2-4mm postganglionic nerves which are unmyelinated. Unmyelinated (post ganglionic) nerves in the penis obviously are not injured during a RP.

Authors’ response: The authors thank the reviewer for this important comment. We have included the “unmyelinated” term because there a number of studies stating like that. In particular, Chauhan et al. (doi: 10.1590/s1677-55382010000300002) use the same term in their study. In fact, Reeves et al. have demonstrated that 90% of nerves in cross-sectional neural areas at prostate base, mid and apex are unmyelinated (doi10.1016/j.urology.2016.08.027). Another study by Alsaid et al (10.1016/j.eururo.2010.04.002) failed to identify myelinated fibers originating from the inferior hypogastric plexus and innervating prostate, seminal vesicles and erectile bodies.

We agree with the reviewer that the use of the term “unmyelinated” may arise some inaccuracy among readers. Thus we have now avoided mentioning the type of the nerve fibers in our text. Nevertheless, it has been proven that the thermal injury of the nerves impacts the postoperative erectile function. Thus, we believe that eliminating that term does not impact our discussion. In addition, a paragraph discussing the nature of the prostate nerve fibers and the injury classifications have been added, lines 259 – 271.

  1. Further, the physiology of nerve recovery timeline is clearly supported by the anatomic descriptions of the long pre-ganglionic myelinated nerve taking 1-2 years to recovery. This of course explains why what appears to be a perfectly preserved nerve after prostatectomy takes months to 2 years to recover. Further, it is also supported by Seddon’s model of nerve injury and recovery and Wallerian degeneration principles.

Authors’ response: The authors thank the reviewer for the comment. This comment is addressed together with the comment number 5.

  1. The authors need to include a clear description of nerve recovery as described by Seddon in the 1940s. There continues to be an embarrassing lack of physiologic knowledge throughout medicine. There is no such word as Neuropraxia. As described by Seddon the word is Neurapraxia (except now as it has mistakenly been “colloquially” accepted for the correct term).

Authors’ response: As suggested by the reviewer a paragraph discussing the types of nerve injuries has been added in the text lines 262 – 271. The word neuropraxia have been corrected throughout the manuscript.

  1. And neurapraxia and more important axonotmesis trauma only occurs in myelinated nerves. Myelinated nerves can suffer these two types of recoverable injury both of which require that the myelinated sheath is intact, i.e. good surgical preservation of the NVBs. Neurapraxic injury is like a concussion and recovers within weeks. The more common and typical recoverable injury is axonotmesis where the (axon) nerve is injured (by heat or traction along the prostate) and then goes through Wallerian degeneration and as long as the sheath was preserved, the axon will grow back through the sheath to the cavernosal bodies at about 1 mm/day and hence the delayed recovery of 9-24 months.

Authors’ response: The authors thank the reviewer for the detailed comment. The information has been now added in the text lines 262 – 271.

  1. In the presentation of thermal injury, there is strong evidence of another school of thought that should be added. Several papers point out that not all thermal injury is the same. In 2002-4 thermal injury occurred when the prostatic vascular pedicles were controlled with bipolar cautery paddles. There are papers and physics that indicate the 20-60 watts of monopolar or bipolar create the same amount of heat energy and potential injury. The major difference between the two is that bipolar paddles when applied to the pedicles occludes blood flow creating desiccating heat and injury. This point was published by Khan F, et.al. J. Endo 2007:21,1195-8.  DOI: 10.1089/end.2007.9908   Subsequently, cut and cauterizing which, unlike the bipolar paddles, preserves adjacent blood flow and has been demonstrated to have the same outcomes as an athermal approach.   Hoffman M, et.al. Eur. Urol. 81(2022),104-09. https://doi.org/10.1016/j.eururo.2021.07.005 

Authors’ response:  The references (42, 43) and respective statements have been now added in the text lines 293-297.

Reviewer 2 Report

This article is a review article with regards to NS techniques in radical prostatectomy and it is basically well-written and informative for the readers of "Cancers".  The reviewer would like to suggest a several issues to be added to the review as below:

1) Please add representative photographs of their own RRPs about each procedure in Fig 2. Those will be helpful for the readers to understand.

2) Endopelvic fascia may not be correct as you presented in Fig 1 and 2.  Please provide both endopelvic fascia and levator ani fascia for the readers to  accurately approach during RRP.

3) Please provide schemes of different NS grading systems.  In order for the readers to understand the differences, the authors should present schemes of Tewari AK et al. (PMID 2191701) and Schatloff O et al. (PMID 22230713).

4) Please provide future aspects which the authors consider.

Author Response

Reviewer 2

1: Please add representative photographs of their own RRPs about each procedure in Fig 2. Those will be helpful for the readers to understand.

Authors’ response: We have now added a new figure showing the key steps of nerve sparing procedure.

2) Endopelvic fascia may not be correct as you presented in Fig 1 and 2.  Please provide both endopelvic fascia and levator ani fascia for the readers to accurately approach during RRP.

Authors’ response: Dear reviewer we cannot find why endopelvic fascia is not correctly presented in Fig 1 and Fig. 2.

 Levator ani fascia is the endopelvic fascia overlying levator ani muscle (which is not presented in both figures). Based on your comment we have expanded Figure 1 laterally as to depict prostatic ani muscle and its fascia. Our understanding of fascial planes is based on the work of Cornu et al 2010 (Fascia surrounding the prostate: clinical and anatomical basis of the nerve-sparing radical prostatectomy. Surg Radiol Anat. 2010 Aug;32(7):663-7).

3) Please provide schemes of different NS grading systems.  In order for the readers to understand the differences, the authors should present schemes of Tewari AK et al. (PMID 2191701) and Schatloff O et al. (PMID 22230713).

Authors’ response: Dear reviewer we do not have the copyrights to use the excellent figures of Tewari and Schatloff and it is not easy to make a completely new set of original illustrations on this. Still, given that both articles have been cited both in the respective table and in the text we believe that readers will have easy access to this really important material you are pointing to.  

4. Please provide future aspects which the authors consider.

Authors’ response: Dear reviewer, as shown different nerve-sparing techniques are associated with improved functional outcomes. As surgery and surgical techniques continuously evolve, there will be more modifications in the future. As for now, our beliefs for nerve preservation are in line with the reported techniques. 

Round 2

Reviewer 2 Report

Authors responded as required.